# *STAT3* Mutation Is Associated with STAT3 Activation in CD30^+^ ALK^−^ ALCL

**DOI:** 10.3390/cancers12030702

**Published:** 2020-03-16

**Authors:** Emma I. Andersson, Oscar Brück, Till Braun, Susanna Mannisto, Leena Saikko, Sonja Lagström, Pekka Ellonen, Sirpa Leppä, Marco Herling, Panu E. Kovanen, Satu Mustjoki

**Affiliations:** 1Hematology Research Unit Helsinki, Department of Hematology, University of Helsinki and Helsinki University Hospital Comprehensive Cancer Center, 00290 Helsinki, Finland; emma.andersson@hus.fi (E.I.A.); oscar.bruck@helsinki.fi (O.B.); 2Translational Immunology Research program and Department of Clinical Chemistry and Hematology, University of Helsinki, 00014 Helsinki, Finland; 3HUSLAB, Laboratory of Genetics, Helsinki University Hospital, 00290 Helsinki, Finland; 4Department I of Internal Medicine, CMMC, CECAD, CIO-ABCD, University of Cologne, 50931 Cologne, Germany; till.braun@uk-koeln.de (T.B.); marco.herling@uk-koeln.de (M.H.); 5Department of Oncology, Comprehensive Cancer Center, Helsinki University Hospital, 00290 Helsinki, Finland; susanna.mannisto@hus.fi (S.M.); sirpa.leppa@hus.fi (S.L.); 6Department of Pathology, HUSLAB, Helsinki University Central Hospital and University of Helsinki, 00290 Helsinki, Finland; leena.saikko@helsinki.fi (L.S.); panu.kovanen@helsinki.fi (P.E.K.); 7Institute for Molecular Medicine Finland (FIMM), University of Helsinki, 00014 Helsinki, Finland; sonja.lagstrom@kreftregisteret.no (S.L.); pekka.ellonen@helsinki.fi (P.E.)

**Keywords:** lymphoma, T-cells, STAT3, RHOA, NGS

## Abstract

Peripheral T-cell lymphomas (PTCL) are a heterogeneous, and often aggressive group of non-Hodgkin lymphomas. Recent advances in the molecular and genetic characterization of PTCLs have helped to delineate differences and similarities between the various subtypes, and the JAK/STAT pathway has been found to play an important oncogenic role. Here, we aimed to characterize the JAK/STAT pathway in PTCL subtypes and investigate whether the activation of the pathway correlates with the frequency of *STAT* gene mutations. Patient samples from AITL (*n* = 30), ALCL (*n* = 21) and PTCL-NOS (*n* = 12) cases were sequenced for *STAT3*, *STAT5B, JAK1, JAK3,* and *RHOA* mutations using amplicon sequencing and stained immunohistochemically for pSTAT3, pMAPK, and pAKT. We discovered *STAT3* mutations in 13% of AITL, 13% of ALK^+^ ALCL, 38% of ALK^−^ ALCL and 17% of PTCL-NOS cases. However, no *STAT5B* mutations were found and *JAK* mutations were only present in ALK^-^ ALCL (15%). Concurrent mutations were found in all subgroups except ALK^+^ ALCL where *STAT3* mutations were always seen alone. High pY-STAT3 expression was observed especially in AITL and ALCL samples. When studying JAK-STAT pathway mutations, pY-STAT3 expression was highest in PTCLs harboring either *JAK1* or *STAT3* mutations and CD30^+^ phenotype representing primarily ALK^−^ ALCLs. Further investigation is needed to elucidate the molecular mechanisms of JAK-STAT pathway activation in PTCL.

## 1. Introduction

Peripheral T-cell lymphomas (PTCLs) form a heterogeneous, uncommon, and often aggressive group of non-Hodgkin’s lymphomas (NHL) representing approximately 10–15% of all new NHL diagnoses [1]. The most prevalent PTLCs are PTCL not otherwise specified (NOS), angioimmunoblastic T-cell lymphoma (AITL), anaplastic lymphoma kinase–negative (ALK^−^ ALCL), and anaplastic lymphoma kinase-positive anaplastic large cell lymphoma (ALK^+^ ALCL) [2,3].

Despite generally favorable response rates to chemotherapy, remissions are often not durable, and as such the natural history of PTCL is characterized by relapses and refractory disease. For this reason, upfront hematopoietic stem cell transplantation (HSCT) is often recommended in first remission for patients who are fit and have chemo-sensitive disease [4]. For patients who are not suitable for transplantation, chances of long-term disease control are very limited, with an average progression-free survival (PFS) of 5.5 months [5]. In this setting, optimal therapeutic approaches remain undefined representing an unmet clinical need.

Recent advances in the molecular and genetic characterization of PTCL have helped to delineate differences and similarities between the various subtypes [6]. Several recurrent mutations have been identified in small subsets of patients potentially enabling more accurate disease classification and guide treatment decisions. In AITL, the three most commonly identified genetic lesions occur in the Tet methylcytosine dioxygenase 2 gene (*TET2*), the Ras homolog gene family, member A (*RHOA*), and the isocitrate dehydrogenase 2 gene (*IDH2*). Besides AITL, *TET2* mutations can be seen with a high frequency also in PTCL-NOS with a T follicular helper cell phenotype [7]. RHOA, a small GTPase participating in T-cell activation and polarization, has recently been found to have a specific G17V mutation in 68% of AITL cases [8], predominantly in the background of *TET2* mutations. The overwhelming majority of *IDH2* mutations in PTCL affect the R172 residue [9]. However, *STAT3* mutations are relatively uncommon (5%) in AITL [10].

In ALK^+^ ALCL, the ALK chimeras activate STAT3, thus maintaining the neoplastic phenotype in ALK^+^ ALCL. In concordance with this, gene expression profiling has revealed a transcriptional gene signature including *ALK*, *TNFRSF8* (*CD30*), *MUC1*, Th17-associated genes (*IL-17A*, *IL-17F*, and *ROR-γ*), a small group of immunoregulatory cytokines and receptors regulating STAT3 pathway [11].

A systematic characterization of the genetic alterations driving ALCL was recently undertaken using sequencing strategies. Activating mutations of *JAK1* and *STAT3* genes were found in 20% of ALK^−^ ALCLs, 38% of which displayed double lesions [12]. As the JAK/STAT pathway has a critical role in hematopoietic development, it is not surprising that it plays, when deregulated, an important oncogenic role in lymphoproliferative malignancies [13]. Many cancers and hematologic malignancies have been associated with the constitutive activation of the STAT family of proteins, which depends on JAK-mediated tyrosine phosphorylation for transcriptional activation [14]. In particular, activated JAK1/STAT3 and JAK2/STAT5 have been shown to facilitate T-cell transformation [15,16]. Activating mutations of *JAK1-3* and *STAT3-5* have also been found in a subset of NK/T-cell lymphomas, non-hepatosplenic gamma-delta T-cell lymphoma, large granular lymphocytic leukemia and T-cell prolymphocytic leukemia [10,17,18,19,20,21,22].

Here, we aimed to further characterize the JAK/STAT pathway in PTCL by investigating whether the activation of the pathway quantified with immunohistochemistry correlates with the frequency of *JAK*, *STAT,* and *RHOA* mutations. In addition, we associated JAK-STAT pathway activation with prognosis and essential clinical and pathological parameters. Finally, we identified novel therapeutic approaches for patients with high STAT3 phosphorylation using Reverse Phase Protein Array and drug sensitivity data from the Broad Institute Cancer Cell Line Encyclopedia (CCLE).

## 2. Results

### 2.1. STAT3 Mutations Are Frequent in PTCLs

As STAT3 is constitutively expressed in both AITLs and ALCLs, we sequenced for *STAT3* and *STAT5B* mutations. The entire *STAT3* gene and the hotspot SH2-domain of *STAT5B* were screened by targeted amplicon sequencing in 63 patients with T-cell lymphoma. We discovered that 13% of AITL and ALK^+^ ALCL cases harbored *STAT3* mutations, while 17% of PTCL-NOS cases were found to harbor *STAT3* mutations (Table 1). The highest frequency of *STAT3* mutations was found in the ALK^−^ ALCL subgroup (38%). The frequency of *STAT3* mutations in the entire patient cohort was 19%. While most mutations were found in the SH2-domain of *STAT3*, we observed one K290T mutation in the coiled coil domain and three patients harbored a P715L mutation in the transactivation domain (Figure 1 and Appendix A). The most prevalent mutation type was the P715L mutation and the E616 deletion, which were identified in three patients each. No *STAT5B* SH2-domain mutations were identified in this cohort. *JAK1* mutations were found in two ALK^−^ ALCL patients (15%). Both mutations (G902R and G1097C) were seen in the protein tyrosine kinase domain of *JAK1*. No *JAK3* mutations were detected in any subgroup.

As *RHOA* mutations are highly prevalent in AITL and have been found to cause activation of the MAPK/PI3K/AKT pathway and T cell receptor signaling, the hotspot mutation G17V was also screened [23,24]. The results showed that these mutations were frequent in AITL (70%) and PTCL-NOS (17%) but remained absent in ALCLs (Table 1). Some cases harbored both *STAT3* and *RHOA* mutations (10%). DNA samples from nodular lymphocyte predominant Hodgkin lymphoma with abundant follicular T helper cells (*n* = 13), were also available for *STAT3/STAT5B* and *RHOA* screening but no mutations were detected in these samples.

### 2.2. STAT3 Phosphorylation Levels Are Highest in ALK^+^ and ALK^−^ ALCL Cases

To characterize JAK-STAT pathway activation, we stained for pY-STAT3, pMAPK, and pAKT in AITL (*n* = 29), ALCL (*n* = 20) and PTCL-NOS (*n* = 12) patients and control lymph nodes (*n* = 4) using immunohistochemistry (Appendix A). The differences between disease groups were significant notably for phosphorylation of pY-STAT3 and recapitulated well previous reports [25]. Further post-hoc analysis showed that pY-STAT3 was shown to be more frequently phosphorylated in AITL (mean = 43.7%) and ALCL (ALK^−^: mean = 62.3%, ALK^+^: mean = 79.7%) patients compared to control lymph nodes (mean 12.6%; Figure 2).

MAPK phosphorylation was significantly elevated in AITL patients (mean 10.5%) followed by ALK^−^ ALCL (mean 6.8%) and PTCL NOS (5.2%) when compared to control lymph node (Figure 2). No significant difference in pAKT levels between PTCL patients and control lymph nodes was observed.

To confirm the findings and study STAT3 activation in other T cell malignancies, we measured STAT3 and pY-STAT3 levels by Western blotting in ALCL, acute T-lymphoblastic leukemia (T-ALL), NK/T-cell lymphoma (NKT), T-cell large granular lymphocyte leukemia (T-LGL), and PTCL cell lines (Figure 3). We confirmed high phosphorylation of pY-STAT3 (Tyr705) in ALCL but not in other T cell malignancies. Total STAT3 was most prominent in PTCL, ALCL and T-ALL cell lines. High JAK3 phosphorylation was observed in SupM2, Karpas299 and TLRB1 (ALCL) as well as NKL (NKT) cell lines and was not associated with pY-STAT3 phosphorylation.

To further investigate the association of protein expression to transcriptome data, we compared the *JAK1, JAK3, AKT*, and *MAPK* RNA level expression with *STAT3* RNA level expression and pSTAT3^Tyr705^ phosphorylation defined with *reverse phase protein arrays* (RPPA) from the CCLE database (Appendix A). Both *STAT3* expression (535 vs. 198 CPM *p* = 0.003, *t*-test) and phosphorylation (3.6 vs. −0.10 RPPA unit, *p* < 0.001) were observed to be superior in ALCL than non-ALCL malignant T-cells as shown in Figure 2. Higher *JAK3* expression was noted in ALCL over non-ALCL lines (364 vs. 53 CPM, *p* < 0.001), and correlated with *STAT3* expression (*r* = 0.60, *p* = 0.01). Moreover, higher *MAPK* expression was detected in ALCL over non-ALCL cell lines (127 vs. 68, *p* = 0.05), which correlated with *STAT3* expression (*r* = 0.65, *p* = 0.006). No correlation between *STAT3* and *JAK1* or *AKT* expression was observed.

### 2.3. STAT3 Phosphorylation Is Associated with JAK1/STAT3 Mutation Status in CD30^+^ ALK^−^ ALCLs

Next, we studied whether *STAT3* mutations would be associated with STAT3, MAPK, and AKT phosphorylation. No association between *STAT3* mutation status and pY-STAT3, pMAPK, or pAKT (%) expression was noted when studying all PTCL subtypes (Figure 4A). No association with pY-STAT3 phosphorylation was noted even after adjusting with tumor proportion (mean pSTAT3 77% vs. 66% in STAT3 mutated vs. non-mutated PTCLs). Most of the mutations were noted in relatively low variant allele frequencies (VAF 5–9%), which might in part explain the lack of correlation. Further evidence for this observation was found from cell line data, as the four ALCL cell lines in Figure 2 exhibit high pY-STAT3 phosphorylation but only one of the cell lines harbored a *JAK/STAT* mutation (Appendix A). The TLBR1 cell line represents ALK^−^ ALCL and harbors a known activating *STAT3* missense mutation S614R in the SH2 domain [26]. The other ALCL cell lines expressed ALK, which explains the constitutive STAT3 activation. Both T-ALL cell lines MOLT4 and Jurkat also harbor missense *STAT3/5B* mutations but conversely, the cell lines did not exhibit any pY-STAT3 phosphorylation. This could be explained by the mutations being localized elsewhere than in the activating SH2-domain.

Some PTCL cases harbored both *STAT3* and *RHOA* mutations (8%). Our results showed that the MAPK phosphorylation degree was higher in patients with *RHOA* mutations than patients without *RHOA* mutations (*RHOA*-positive mean = 12.4% vs. *RHOA*-negative mean 5.7%, *p* = 0.005, t-test; Figure 4B). Yet, *RHOA* mutation was not observed to be associated with STAT3 or AKT phosphorylation.

Phosphorylated STAT3 has been shown to induce CD30 expression in PTCL [27]. In addition, *STAT3* mutations are more frequent in CD30^+^ than CD30^−^ T-cell lymphomas [28]. Therefore, we sought to investigate the interaction between mutations in the JAK-STAT pathway, CD30 phenotype and pY-STAT3 phosphorylation. Positive CD30 tumor phenotype was associated with higher STAT3 and AKT phosphorylation, but not with pMAPK (Appendix A). Interestingly, pSTAT3 expression was highest in samples with CD30^+^ tumors and simultaneous mutation in *STAT3* or *JAK1* (Figure 4C). This phenotype was represented primarily by the ALK^−^ ALCL subtype (*p* < 0.001, chi^2^ test, Figure 4D), and their pSTAT3 expression (mean 68.7%) to stand above the mean of ALK^−^ (62.3%) but below ALK^+^ ALCL (79.7%). We observed that 50% (6/12) of CD30^+^ ALK^−^ ALCLs harbored a mutation in *JAK1* or *STAT3*. Higher *JAK1/STAT3* VAF trended for higher pY-STAT3 phosphorylation but the result remained non-significant possibly due to sample size (81.3% vs. 61.1% mean pSTAT3 expression in tumors with high (*n* = 4) vs. low (*n* = 4) *JAK1/STAT3* VAF *p* = 0.22, *t*-test). Unexpectedly, lowest STAT3 phosphorylation was observed in CD30^−^ patients with a *STAT3* gene mutation. Interestingly, all three in-frame E616 deletions were found in CD30^−^ AITL cases.

EBV status has been shown to be associated with STAT3 activation in B cell lymphomas [29]. No associations between STAT3, MAPK, and AKT phosphorylation and EBV^+^ tumors could be observed in our PTCL cohort (Appendix A).

As fixation rate differs in samples of various origin and size, we investigated their association with STAT3, MAPK, and AKT phosphorylation. The sample cohort included 9 biopsy samples and 52 excised tumors. Biopsy samples were associated with higher pY-STAT3 but not with MAPK or AKT phosphorylation (Appendix A). Moreover, correlation between pY-STAT3 and sample size remained non-significant (Appendix A). Therefore, we hypothesize the difference to be due primarily to biological rather than technical reasons. Except for four samples, the sample cohort consisted of lymph node tissue samples. AKT phosphorylation was higher in lymph node samples, while no difference was observed between pY-STAT3 and MAPK phosphorylation levels (Appendix A).

To further elucidate the spatial localization of STAT3 phosphorylation, 6 T-cell lymphoma samples (5 AITL, and 1 ALCL) and 2 NLPHL samples were stained for pY-STAT3, PD1, CD4, and CD8 using multiplex immunohistochemistry. Of the 5 AITL samples, two harbored *STAT3* mutations at 5% and 7% VAF, respectively. We found that the staining pattern or level of pY-STAT3 phosphorylation was not associated with *STAT3* mutation status (Figure 5).

### 2.4. High pY-STAT3 Expression Is Associated with CD3^−^ CD5^−^ CD7^−^ CD30^+^ Immunophenotype Common to ALCL

To understand the association of JAK-STAT signaling pathway activity with clinical determinants, we combined pY-STAT3, pMAPK, and pAKT proportion defined by IHC with information on PTCL disease histology, stage, peripheral blood lactate dehydrogenase (LD) level at diagnosis, survival status, *STAT3* and *RHOA* mutation status (Figure 6). We noted that pY-STAT3, pMAPK, and pAKT expression did not correlate with each other. High pY-STAT3 expression was noted to be associated with ALCL histology as presented before (Figure 2). No association with survival status (*p* = 0.17, *t*-test), disease stage (*p* = 0.40, one-way ANOVA), LD level (*r* = -0.13, *p* = 0.31, Pearson correlation), *STAT3* (*p* = 0.56, *t*-test) or *RHOA* mutation status (*p* = 0.11, *t*-test) was noted.

To correlate the expression of the JAK-STAT signaling pathway with immunohistochemical and serological factors of disease pathology, we investigated EBV seropositivity, and expression of immunophenotype markers as defined with IHC (Figure 6). Lower pY-STAT3 expression was associated with negative CD3 (37.7% and 75.4% mean pSTAT3 expression, *p* < 0.001, *t*-test)*,* CD5 (42.8% and 68.4%, *p* = 0.03, *t*-test)*,* CD7 (45.2% and 64.6%, *p* = 0.01) and positive CD30 phenotype (Appendix A) designating ALCL immunophenotype as reported in Figure 2 [30]. No association with EBV seropositivity (Appendix A), CD4 (*p* = 0.86, *t*-test) or CD8 (*p* = 0.35, *t*-test) immunophenotype nor MIB proliferation index (*r* = 0.20, *p* = 0.19, Pearson correlation) was noted.

### 2.5. Novel Potential Inhibitors of STAT3 Activation

In myeloproliferative neoplasms the JAK-STAT signaling pathway can be inhibited with the JAK1/JAK2 inhibitor ruxolitinib and in rheumatoid arthritis with less selective JAK inhibitors such as tofacitinib and baricitinib [31]. Several other pharmacological compounds affecting the JAK-STAT pathway are currently investigated in clinical trials [31]. We hypothesized that drugs investigated for other indications might induce differential sensitivity in cancer cell lines with active JAK-STAT signaling. By comparing cytotoxicity measured by the area under the curve (AUC) of 265 investigational and accepted compounds in pSTAT3 high (*n* = 27) and low (*n* = 583) cancer cell lines reported in the Sanger GDSC and CCLE databases respectively, we identified the JAK-STAT inhibitor ruxolitinib to be more potent in pSTAT3 high cell lines (Figure 7). Interestingly, we discovered also the cell cycle checkpoint kinase Chk1/2 inhibitor AZD7762, the poly ADP ribose polymerase (PARP) inhibitor talazoparib and the nucleoside analog gemcitabine to exhibit most sensitivity in pSTAT3 cell lines.

## 3. Discussion

PTCL patients face unfavorable prognosis as chemotherapy results in poor 5-year overall survival rates [32]. Targeted treatment options only exist for ALK^+^ ALCL, but are urgently needed for other PTCL entities. Using targeted sequencing, we demonstrated that *STAT3* alterations are prevalent in all PTCL subgroups. We discovered that 13% of AITL and ALK^+^ ALCL cases and 17% of PTCL-NOS cases harbored *STAT3* mutations. The highest prevalence of *STAT3* mutations was seen in ALK^−^ ALCL (38%). The mutations were found mostly in the SH2-domain of *STAT3* but also in the coiled coil (K290T) and transactivation domains (P715L). While we observed no association between overall *STAT3* mutation and pY-STAT3 phosphorylation, pY-STAT3 phosphorylation was most elevated in PTCL patients with combined CD30^+^ tumor phenotype and mutations in either *STAT3* or *JAK1* representing primarily ALK^−^ ALCLs and suggesting heterogenous mechanisms of STAT3 activation. Unexpectedly, PTCLs with CD30^-^ phenotype and *STAT3* mutation were associated with lowest pY-STAT3 phosphorylation. As CD30 transcription is regulated by pSTAT3, the result might be explained by non-activating mutations or interference in the JAK-STAT pathway [27].

*JAK/STAT* mutations have been previously reported in PTCL-NOS, ALCL, and AITL [12,33,34], supporting the results of our study. The frequency of *JAK/STAT* mutations in different cohorts of PTCL has been reported to be around 20% which compares well to the mutation frequency seen in this study. In the whole cohort one of the most prevalent *STAT3* mutation was an in-frame deletion of E616, which was identified in three AITL patients. The E616 deletion has been shown to induce myeloid malignancy in a mouse bone marrow transplantation model [19]. The mutation has been observed previously in AITL, but also in adult T-cell lymphoma [10,35]. The *STAT3* mutation P715L observed in two ALK^−^ ALCLs and one PTCL-NOS case has been observed in NKTL previously [36].

Some AITL and PTCL-NOS cases harbored concurrent *STAT3* and *RHOA* mutations (3/30 AITL and 1/12 PTCL-NOS). The incidence of *RHOA* mutations in our AITL cohort was 70%, which is somewhat higher than the incidence seen in previous AITL cohorts (68% and 53%) [8,23]. In PTCL-NOS the mutation incidence was 17% (2/12). Our results also showed that the MAPK phosphorylation degree correlated with *RHOA* mutations in our PTCL cohort.

Although this study included limited number of patients representing various PTCLs subtypes, our results suggest that pY-STAT3 is frequently constitutively phosphorylated in PTCL, especially in ALCLs and AITLs, but this was associated with *STAT3* mutations in only a fraction of PTCL samples. Previous studies have reported mechanisms underlying the constitutive activation of STAT3 independent of *STAT3* mutations. Gain-of–function mutations involving *JAK* genes have been implicated in activating STAT3 and contributing to the pathogenesis of hematologic malignancies and solid tumors [37,38]. Various cytokines (e.g., IL-6, IL-10, and IL-11) that are released into the microenvironment by tumor cells have also been shown to activate STAT3 [39]. Another mechanism of STAT3 activation is possible through the deregulation of suppressors of the JAK/STAT pathway such as SOCS3 [40] leading to constitutive STAT activation and oncogenesis. In addition, further studies are needed to combine any association with JAK-STAT pathway activation and expression of essential transcription factors such as GATA3 and Tbet, which define new subclasses of PTCL-NOS with prognostic consequences, where GATA3 signatures are associated with worse prognosis [41,42].

In our cohort high pSTAT3 expression correlated with the CD3^−^ CD5^−^ CD7^−^ CD30^+^ immunophenotype typical for ALCL. No association with neither immunophenotype markers nor essential clinical variables such as survival status, disease stage, MIB proliferation index or LD level was seen.

By comparing cytotoxicity measured by the area under the curve (AUC) of 265 investigational and accepted compounds in pSTAT3 high and low cancer cell lines, we identified novel potential inhibitors of STAT3 activation. The JAK-inhibitor ruxolitinib was among the top candidates. Ruxolitinib, a selective JAK1/2 inhibitor approved by the FDA for myeloproliferative neoplasms, is being studied in relapsed B-cell lymphoma and PTCL (NCT01431209).

Among compounds exhibiting most sensitivity in pSTAT3 high cell lines were the cell cycle checkpoint kinase Chk1/2 inhibitor AZD7762, the poly ADP ribose polymerase (PARP) inhibitor talazoparib, and the nucleoside analog gemcitabine. In phase II studies, gemcitabine monotherapy has shown activity against T-cell lymphomas. Zinzani et al. reported a 70% response rate in a phase II study of 44 pretreated patients with mycosis fungoides or cutaneous PTCL-unspecified [43]. In another phase II study, gemcitabine was given in combination with romidepsin to relapsed/refractory PTCL patients but the synergy observed in preclinical phase did not improve clinical outcomes [44]. Moreover, gemcitabine has been tested in combination with cisplatin and methylprednisolone as both upfront treatment and in relapsed/refractory PTCL, but did not significantly improve OS [45,46].

## 4. Materials and Methods

### 4.1. Patients and Cell Lines

Samples from 63 patients with PTCL were collected. Of these samples 30 were AITL, 12 PTCL-NOS and 21 ALCL (13 ALK^−^ and 8 ALK^+^). Four control samples from normal lymph nodes were also collected. The study was undertaken in compliance with the principles of the Helsinki Declaration and was approved by the ethics committee in the Helsinki University Hospital (Finland). As the samples studied were older diagnostic samples, the ethics committee required no written informed consent (TEO 5326/04/046/06, Valvira 9115/05.01.00.06/2011, HUS 302/E0/2006, HUS/1230/2017). The clinical and pathological characteristics of our study objects are summarized in Appendix A.

### 4.2. Cell Lines and Cell Culture Conditions

The T-cell lines were cultured in RMPI-1640 medium including L-glutamine (Gibco, Grand Island, NY, USA) supplemented with 10% (L82, Karpas299, TLRB1, Molt4, Jurkat, and NKL cell line) or 20% (SupM2, MTA, MOTN1, and SMZ1 cell line) fetal bovine serum (FBS, Gibco) and Penicillin/Streptomycin (100 U/l, Gibco). IL2 (10ng/mL, MedChemExpress, Monmouth Junction, NJ, USA) was added for culturing of TLRB1 and NKL cells. Suspension cells were maintained at a density of 2.0–8.0 × 10^5^. Culturing was done in an incubator at 37 °C and 5% CO_2_ with 90% humidity.

### 4.3. Sample Preparation

According to standard clinical procedure, fresh tissue samples consisting mostly of lymph nodes (LN) were fixated in formalin and embedded in paraffin (FFPE) in the central pathology laboratory of Helsinki University Hospital (HUCH), Finland. Consequently, we cut 3.5 µm whole-tissue sections on Superfrost objective slides (Kindler O Gmbh, Freiburg, Germany).

### 4.4. DNA Extraction

DNA was extracted from FFPE tissue blocks. Ten 10 μm sections were cut with standard microtome (Leica SM2000 R Sliding Microtome, Wetzlar, Germany) using disposable blades. Excess paraffin was trimmed off and the sections were collected into sterile 1.5 mL microcentrifuge tubes. The sections were then deparaffinized with three pre-warmed (55 C) xylene washes followed by 95%, 75%, and 50% ethanol rinses. The tissue pellets were dried briefly at 37 °C to remove traces of ethanol. The pellets were then digested with 20 µL proteinase K (20 mg/mL proteinase K, Roche Diagnostics GmbH, (Mannheim, Germany) and 180 µL digestion buffer (10 mM Tris-HCl, pH 8.0 100 mM EDTA, pH 8.0 50 mM NaCl, and 0.5% SDS). Samples were incubated at 55 °C with mild agitation for 3 to 72 h. Fresh Proteinase K was added every 24 h followed by heat inactivation at 90 °C for 1 h after the tissue had been fully dissolved. For phenol-chloroform extraction, equal volume of phenol (Amresco, Solon, OH, USA) was added and vortexed. After spinning for 3 min at 14,000 rpm, the aqueous layer was transferred to a new tube. Then, an equal volume of phenol-chloroform-isoamyl alcohol (25:25:1) (Invitrogen, Carlsbad, CA, USA) was added, and the solution was vortexed and spun for 5 min at 14,000 rpm in a microcentrifuge. The aqueous layer was again transferred to a new tube and treated with RNase A at 100 μg/mL for 1 h at 37 °C. To remove any remaining RNase A phenol and phenol-chloroform-isoamyl alcohol steps were repeated. The aqueous layer was transferred to a new tube and the DNA was precipitated with 1/10 volume of 3 M sodium acetate and 1 volume of isopropanol. After thorough mixing, the solution was placed in a freezer for 30 min then spun at 14,000 rpm at 4 °C in a microcentrifuge for 10 min. The supernatant was discarded, and the pellet was washed with 1 mL 70% cold ethanol and spun at 14,000 rpm for 10 min at 4 °C. The supernatant was carefully discarded, and the pellet dried and finally suspended with 50 µL dH_2_O.

### 4.5. Targeted Deep Amplicon Sequencing

Locus-specific primers were designed for *STAT3, STAT5B, JAK1, JAK3,* and *RHOA* mutation hotspots using Primer3 with user-defined parameters (http://primer3.wi.mit.edu/). After designing the locus-specific primer sequences (Appendix A), sequence tails corresponding to the Illumina adapter sequences, were added to the 5′ end of the forward and reverse locus-specific primers, respectively. All oligonucleotides were ordered from Sigma-Aldrich (St. Louis, MO, USA). Deep targeted amplicon sequencing of known recurrent somatic mutations in the genes *STAT3, STAT5B*, *JAK1, JAK3*, and *RHOA* was performed with the Illumina MiSeq platform (San Diego, CA, USA)as previously described [47]. The data were analyzed with a bioinformatics pipeline, which is based on calling of variants with certain count/frequency of reads and filtering out false positives using the estimated error rate and quality data of amplicon reads.

### 4.6. Immunohistochemistry

#### 4.6.1. Single Color Immunohistochemistry

Tissue sections were deparaffinized, rehydrated in graded ethanol series and then subjected to heat-induced epitope retrieval in Tris-HCl (10 mM Tris, pH 8.5, 25 min; for Phospho-MAPK and Phospho-AKT) or in EDTA buffer (10 mM EDTA, pH 8.0, 20 min; for Phospho-Stat3 and -5) in a Pre-Treatment module (DAKO/Agilent Technologies, Waltham, MA, USA). Samples were stained using LabVision autostainer (Thermo Fisher Scientific, Fremont, CA, USA). The endogenous peroxidase activity was blocked with Peroxidase-Blocking Solution (Dako/Agilent Tehnologies). The samples were treated with primary antibody for 1 h at RT. The antibodies used were rabbit monoclonal antibodies from Cell Signaling (Leiden, the Netherlands): clone Tyr705 for pY-STAT3 (1:300), clone Thr202/Tyr204 for Phospho-p44/42MAPK (1:300) and Thr450 for Phospho-AKT (1:100). BrightVision polymerisation technology was utilized to prepare polymeric HRP-linker antibody conjugates (ImmunoLogic, Duiven, the Netherlands) and 3,3-diaminobenzidine (DAB) was used as chromogen. Sections were counterstained with Mayer’s hematoxylin (Dako/Agilent Tehnologies) and mounted with Eukitt (Honeywell Fluka, Frankfurt, Germany).

Protein expression was defined by visual examination. Five representative 400×-magnified high-power field-of-view (FOV) were selected, and protein expression (%) was estimated as the proportion of positive cells of 100 cells analyzed by FOV. Finally, the mean proportion of positive cells was calculated and used for statistical analyses.

#### 4.6.2. Multiplex Immunohistochemistry

All phases were performed in room temperature (if not otherwise specified) and protein blocking as well as antibody incubations were performed in a humid chamber. The slides were washed with 0.1% Tween-20 (Thermo Fisher Scientific) diluted in 10 mM Tris-HCL buffered saline pH 7.4 (TBS) three times after peroxide block, each antibody staining, and fluorochrome reaction. Antibodies were tested with normal lymph node and pY-STAT3 was validated with tissue samples from a patient with large granular lymphocyte leukemia (Appendix A).

The slides were deparaffinized in xylene and rehydrated with graded series of ethanol and H2O. HIER was performed in 10 mM Tris-HCl - 1 mM EDTA buffer (pH 9) in +99 °C for 20 min (PT Module, Thermo Fisher Scientific). The peroxide activity was then blocked in 0.9% H2O2 solution for 15 min, and 10% normal goat serum was subsequently applied (TBS-NGS) for 15 min. We applied the primary antibody anti-PD1 (1:5000; clone PDCD1; LsBio, Seattle, WA, USA) diluted in TBS-NGS for 1 h 45 min, and then anti-mouse horseradish peroxidase (HRP) conjugated secondary antibodies (Immunologic) diluted 1:2 in washing buffer for 45 min. Then, we applied tyramide signal amplification (TSA) Alexa Fluor 488 (PerkinElmer, Boston, MA, USA) diluted 1:50 in TBS on the slides for 10 min. In order to multiplex antibodies, we denatured the primary antibody and quenched the enzymatic activity of HRP by repeating HIER as well as performing peroxide and protein block similarly as above. We applied the primary antibody against pY-STAT3 (Tyr705; 1:2500; clone D3A7; CellSignaling) overnight in +4 °C, anti-mouse HRP-conjugated secondary antibody diluted 1:5 in washing buffer for 45 min, and TSA Alexa Fluor 555 (PerkinElmer) for 10 min. We repeated HIER, peroxide block and protein block as above. Next, we applied anti-CD4 (1:150; clone EPR6855; Abcam, Cambridge, UK) diluted in TBS-NGS for 3 h and later AlexaFluor647-conjugated secondary antibody (1:300; Thermo Fisher Scientific) and Hoechst 33,342 (1 µg/mL; Sigma-Aldrich) counterstain diluted in washing buffer for 45 min. Finally, we applied ProLong Gold (Thermo Fisher Scientific) to mount the slides.

The tissue slides were scanned with 20x magnification with the Panoramic P250 Flash II whole slide scanner (3DHistech, Budapest, Hungary). We used DAPI, FITC, Cy3, and Cy5 filters and visually-optimized exposure times.

### 4.7. Western Blot

Immunoblots were performed on whole-cell protein lysates according to standard techniques. All primary antibodies (pJAK3^Tyr980/981^, clone D44E3; JAK3, clone D7B12; pY-STAT3^Tyr705^, clone D3A7; STAT3, clone 124H6; beta-ACTIN, clone 13E5) were ordered from Cell Signaling Technology and used at 1:1000 dilutions. Anti-mouse and anti-rabbit, both from Dianova, were used as secondary HRP-coupled antibodies in a 1:5000 dilution. Development of the immunoblots was performed by using Western Bright ECL (Advansta, Menlo Park, CA, USA). The original blots and densitometry readings of each band is available in the Appendix A.

### 4.8. CCLE and DepMap Data

All CCLE pre-processed data were downloaded from the CCLE data portal (https://portals.broadinstitute.org/ccle/data). RPPA data dated 3 October2018, RNA-seq gene count data dated 29 September 2018, cell line annotation dated 26 December 2018, and mutation data dated 18 July 2018 were used. Gene counts were converted to CPM values using the edgeR and the TMM normalization method without log transformation. The cut-off for high vs. low STAT3 phosphorylation was set to pSTAT3^Tyr705^ = 1 (RPPA unit) based on the shape of the pSTAT3^Tyr705^ frequency histogram.

Area under the curve (AUC) drug sensitivity data annotated as “Sanger GDSC v17.3” and dated March 2018 were downloaded from the Cancer Dependency Map portal (https://depmap.org/portal/download/). Drugs not characterized in both pSTAT3 high and low cell line groups were removed from the analysis. Cell lines with no pSTAT3 RPPA data were removed from the analysis.

### 4.9. Statistical Analysis

Comparison of continuous variables between two groups was computed with unpaired *t*-test and between multiple groups with one-way ANOVA and post-hoc correction between disease groups and healthy controls.

Overall survival was defined as the time from diagnosis to death or last follow-up. Patients alive at the last follow-up date were censored. The log-rank test was used to compute Cox proportional-hazards.

Differential drug sensitivity was computed using t-test between pSTAT3 high (*n* = 27) and low (*n* = 583) cancer cell lines. The fold change in drug sensitivity was defined as the mean AUC ratio in pSTAT3 high-to-low cell lines. Drugs with a fold change (FC) less than 1.0 are more sensitive in pSTAT3 high cell lines.

R 3.5.1. [48] and Prism 6.0 was used for statistical analyses. R packages edgeR 3.24.0, forestplot 1.7.2, survminer 0.4.3, ggplot2 3.2.1, and ComplexHeatmap 2.3.2 were used.

## 5. Conclusions

In summary, this study discovered that pY-STAT3 phosphorylation was associated with both ALK^+^ ALCLs and *JAK1*/*STAT3* mutated CD30^+^ ALK^−^ ALCLs suggesting a subgroup potentially benefitting of JAK/STAT targeted therapy.

## Figures and Tables

**Figure 1 cancers-12-00702-f001:**
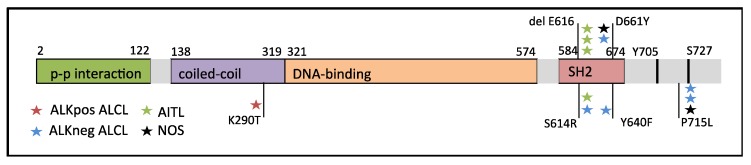
*STAT3* structure and mutation sites in ALCL, AITL, and PTCL-NOS patients of the study cohort.

**Figure 2 cancers-12-00702-f002:**
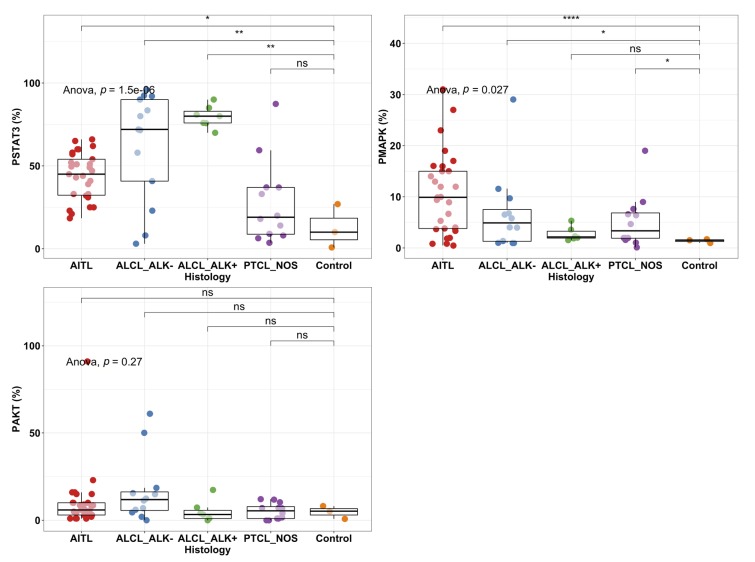
Phosphorylation percentage of pY-STAT3, MAPK, and AKT in PTCL subgroups and healthy controls (*n* = 2–3) analyzed by immunohistochemistry. Individual dots represent the proportion of positive cells for pSTAT3, pMAPK, and pAKT. Boxplots represent the median and interquartile range of protein phosphorylation levels. The antibodies used were specific for Tyr705 (pStat3), Thr202/Tyr204 (pMAPK), and Thr450 (pAKT). Comparison was analyzed with ANOVA and post-hoc tests between disease groups and healthy controls. *: *p* < 0.05; **: *p* < 0.01, ****: *p* < 0.0001; ns: not significant.

**Figure 3 cancers-12-00702-f003:**
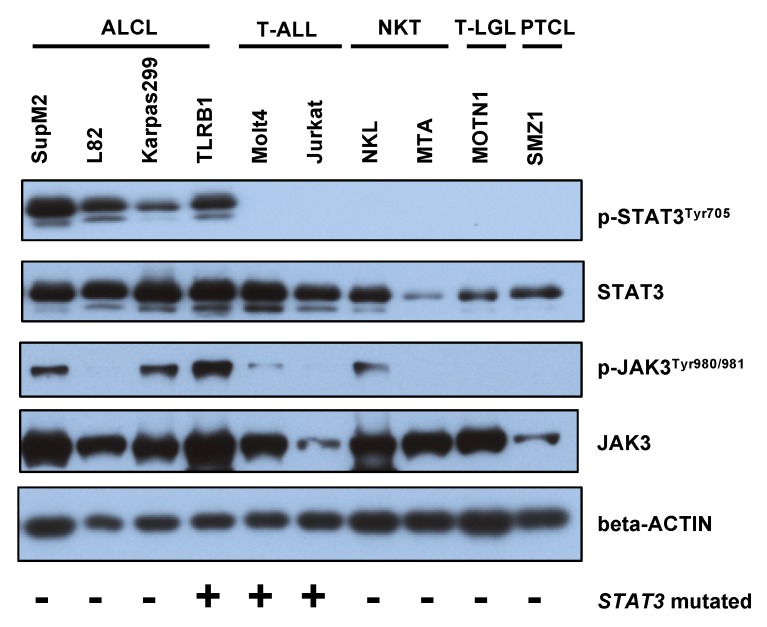
Western blot of p-STAT3 (Tyr705), STAT3, p-JAK3 (Tyr980/981), and JAK3 was characterized in T cell malignancy cell lines. Beta-ACTIN is used as a reference protein.

**Figure 4 cancers-12-00702-f004:**
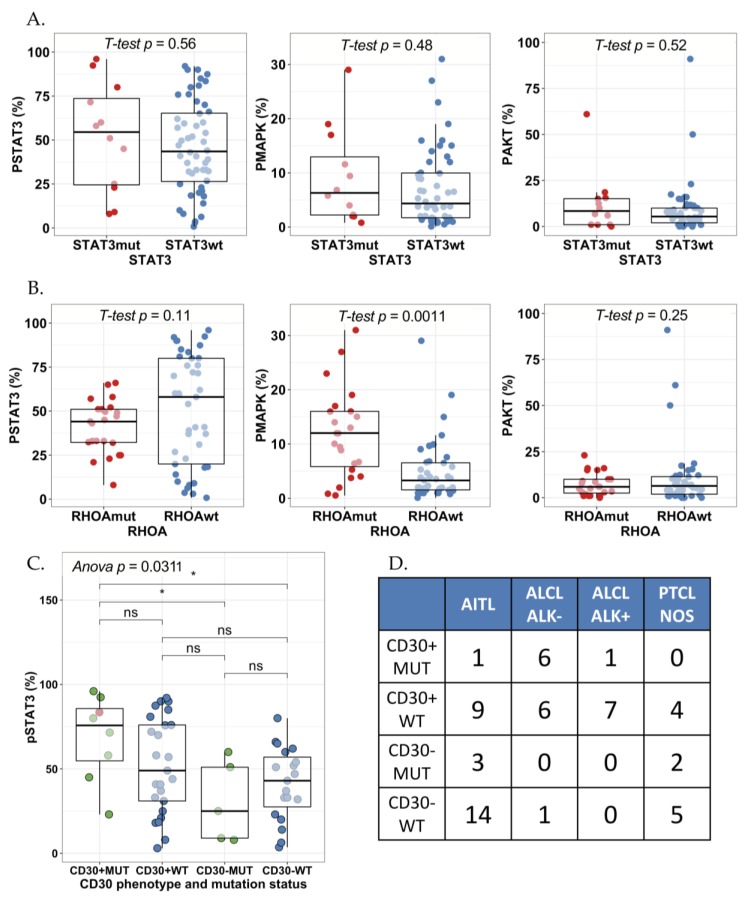
Comparison of the phosphorylation level of (**A**) STAT3, MAPK, and AKT by *STAT3* mutation status and by (**B**) *RHOA* mutation status (*t*-test). The phosphorylation level of pStat3 (Tyr705), pMAPK (Thr202/Tyr204) and pAKT (Thr450) was measured by immunohistochemistry. (**C**) Combinatory effect of CD30 phenotype and *JAK1* or *STAT3* mutation status on pSTAT3 expression was studied with one-way ANOVA and post-hoc correction. Green dots correspond to patients with *STAT3* mutations, the red dot correspond to a patient with *JAK1* mutation and blue dots correspond to samples without mutations in *JAK1* or *STAT3*. (**D**) Frequency table of patients by CD30 phenotype and *JAK1* or *STAT3* mutation status (rows) and different PTCL histologies (columns). *: *p* < 0.05; ns: not significant.

**Figure 5 cancers-12-00702-f005:**
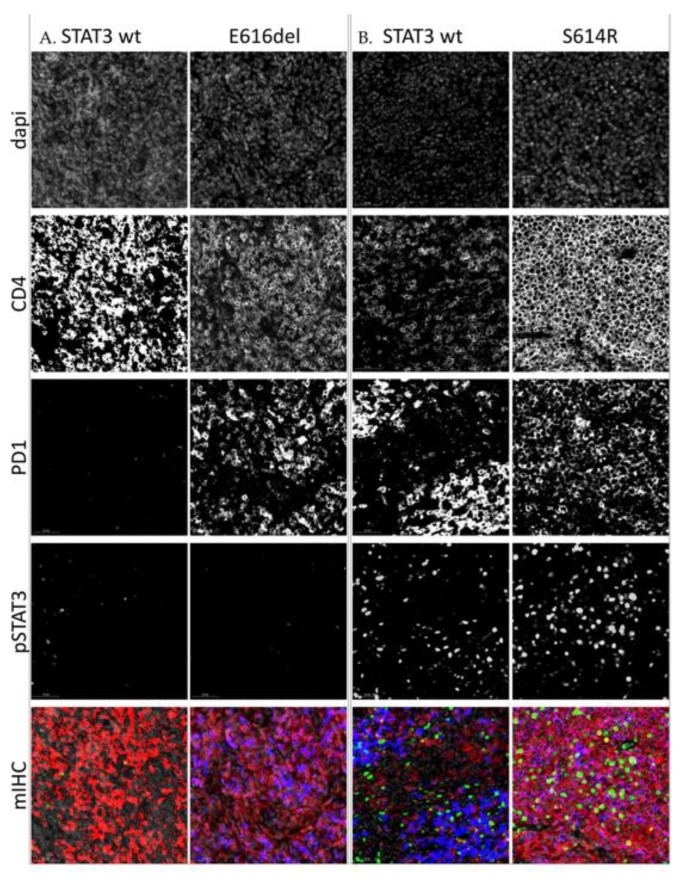
IHC stainings of four angioimmunoblastic T-cell lymphoma (AITL) lymph node samples. (**A**). Samples containing low amount of pSTAT3. (**B**). Samples containing high amounts of pSTAT3. Each slide has been digitalized with a 20× lens and similar exposure time per channel. Images have been further magnified 30× for visualization purposes. The first four rows of images represent single stainings of dapi counterstain, CD4, PD1, and pSTAT3 (Tyr705), respectively, and the last figure their composite image. Staining: red = CD4, blue = PD1, green = pSTAT3.

**Figure 6 cancers-12-00702-f006:**
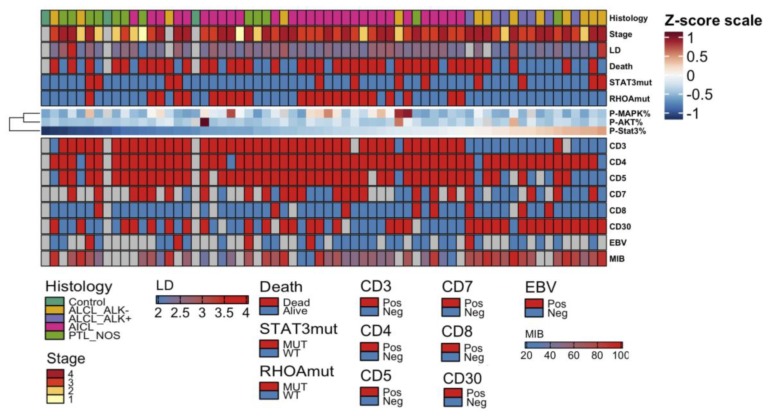
Heatmap visualizing the quantity of pY-STAT3, pMAPK, and pAKT and their association with clinicopathological parameters. The amount of phosphorylated STAT3, MAPK, and AKT have been median-centered and max-scaled, and organized columnwise by pY-STAT3 quantity and rowwise by hierarchical clustering using Spearman correlation distance and Ward linkage (ward.D2) method. Red color denotes higher and blue color lower proportion. Clinical parameters denoting disease histology, disease stage, log10-transformed lactate dehydrogenase (LD) level, survival status, and STAT3 and RHOA mutation status are added as annotations over the heatmap. CD3, CD4, CD5, CD7, CD8, and CD30 immunophenotype status classified as positive or negative, EBV seropositivity and the MIB proliferation index (%) are presented below the heatmap.

**Figure 7 cancers-12-00702-f007:**
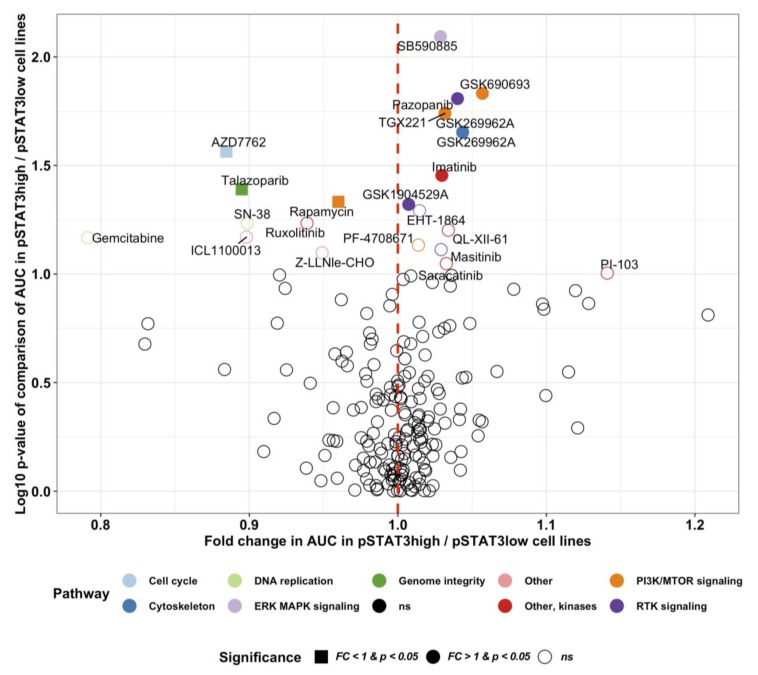
Differential drug sensitivity in pSTAT3 high (*n* = 27) and low (*n* = 583) cancer cell lines. The pSTAT3 phosphorylation level has been quantified with reverse phase protein array (RPPA, cut-off pSTAT3^Tyr705^ = 1). The drug sensitivity data represent the area under the curve (AUC) response of 265 different phamacological compounds, and has been retrieved from the Sanger GDSC dataset. The RPPA data are derived from the Broad Institute the Cancer Cell Line Encyclopedia (CCLE) database. Differential drug sensitivity has been computed with multiple T-tests, and the fold change defined as the mean AUC ratio in pSTAT3 high-to-low cell lines. Drugs with a fold change (FC) less than 1.0 are more sensitive in pSTAT3 high cell lines. The target pathway of the top pharmacological hits have been color-labeled.

**Table 1 cancers-12-00702-t001:** *STAT3, JAK1/3,* and *RHOA* mutation frequencies in PTCLs.

PTCL Subtype	*n*	*STAT3* Mutation Frequency	*JAK1/3* Mutation Frequency	*RHOA* Mutation Frequency	Co-Occuring Mutations
AITL	30	13%	0%	70%	10%
NOS	12	17%	0%	17%	8%
ALK^+^ ALCL	8	13%	0%	0%	None
ALK^−^ ALCL	13	38%	15%	0%	8%
All	63	19%	3%	37%	8%

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
