# Peer review of "STAT3* Mutation Is Associated with STAT3 Activation in CD30^+^ ALK^−^ ALCL"

_cancers, 2020, doi:10.3390/cancers12030702_

Round 1

Reviewer 1 Report

  • STAT3 mutations being relatively common in TCLs is fairly well known
  • pSTAT3 IHC
    • pSTAT3 IHC can be too sensitive to detect increases in phosphorylation, ie normal tissue frequently positive so unclear if pSTAT3 IHC actually showing increased phosphorylation/activation, though findings specifically in ALK- ALCL and not in other PTCL may support this
  • High pSTAT3 expression w/ immunophenotype
    • That is generally the phenotype of ALCL anyway? Basically this says that high pSTAT3 is associated with ALCL, which is redundant
  • Title is wildly discrepant with lines 156-159 which basically state the opposite of the title

Author Response

STAT3 mutations being relatively common in TCLs is fairly well known

# We thank Reviewer 1 for additional comments. We agree that STAT3 is commonly mutated in PTCLs, notably in ALK- ALCL, while lower frequencies have been reported for AITL. This has been discussed in the manuscript in the Introduction (lines 66-67 and 72-74).

pSTAT3 IHC. pSTAT3 IHC can be too sensitive to detect increases in phosphorylation, ie normal tissue frequently positive so unclear if pSTAT3 IHC actually showing increased phosphorylation/activation, though findings specifically in ALK- ALCL and not in other PTCL may support this.

# We observed in Figure 2A that STAT3 phosphorylation is highest in ALK+ and ALK- ALCL patients. As this follows well with previous reports referenced in line 121, we suggest IHC to be suitable for comparing STAT3 phosphorylation in these samples.

High pSTAT3 expression w/ immunophenotype. That is generally the phenotype of ALCL anyway? Basically this says that high pSTAT3 is associated with ALCL, which is redundant.

# We agree with the remark by Reviewer 1. The Figure 6 heatmap visualizes clinical and histopathological variables associated with STAT3 phosphorylation. Individual immunophenotype markers noted to be enriched with high STAT3 phosphorylation represents the ALCL histology as previously observed in Figure 2A, and this has been reported in line 237.

Title is wildly discrepant with lines 156-159 which basically state the opposite of the title.

# We have clarified in line 157 that no association with STAT3 mutation status and pSTAT3 expression was noted when studying all PTCL subtypes as opposed to the ALCL- subtype where these two variables were observed to correlate.

Reviewer 2 Report

The authors improved their paper by adding cases and markers (CD30!), which finally changed the major conclusions drawn. I still have some reservations.

  1. Last time I wrote: "Gene names must be italicized throughout; this is not the case in 80% of the instances." and the authors state now: "# We have reviewed the manuscript and corrected text formatting for gene names." This is completely not the case here: on lines 93, 100, 102, 155, 157, 159, 172, 183, 299, 302 (to name only a few) still all listed genes such as STAT3, JAK1, RHOA and JAK3 are still not italicized. This is the full responsibility of the authors to be as precise as possible, and I would not recommend acceptance prior to that the authors have done their job to correct the paper! I do not need statements, I need to see progress in that consideration.
  2. The sentence on lines 180-182 sounds dull and must be rephrased.
  3. I have reservations respecting figure 2, the text referring to figure 2 and the potential cut-off scores used to designate a case positive for a phosphorylated marker. It is not clear if e.g. the pSTAT3 expression results refer to cases or percentage positive cells. In the materials and methods section, it is written that means have been calculated, but it is not clear if cut-off scores have been used and how they have been assessed.

Author Response

Last time I wrote: "Gene names must be italicized throughout; this is not the case in 80% of the instances." and the authors state now: "# We have reviewed the manuscript and corrected text formatting for gene names." This is completely not the case here: on lines 93, 100, 102, 155, 157, 159, 172, 183, 299, 302 (to name only a few) still all listed genes such as STAT3, JAK1, RHOA and JAK3 are still not italicized. This is the full responsibility of the authors to be as precise as possible, and I would not recommend acceptance prior to that the authors have done their job to correct the paper! I do not need statements, I need to see progress in that consideration.

# We thank for constructive criticism from Reviewer #2. We regret errors in formatting but emphasize that these parts were correctly formatted, and the word files converted to pdf in our computer before submitting the manuscript. We suspect an error to have occurred with the manuscript formatting after submission. We have revised the manuscript for gene name italicization and will discuss with the journal editor to ensure the correct formatting is conserved.

The sentence on lines 180-182 sounds dull and must be rephrased.

# We agree the sentence to have been tedious to interpret and have clarified it.

I have reservations respecting figure 2, the text referring to figure 2 and the potential cut-off scores used to designate a case positive for a phosphorylated marker. It is not clear if e.g. the pSTAT3 expression results refer to cases or percentage positive cells. In the materials and methods section, it is written that means have been calculated, but it is not clear if cut-off scores have been used and how they have been assessed.

# Figure 2 represents the proportion of positive cells for pSTAT3, pMAPK and pAKT per sample. Marker positivity has been assessed by visual inspection as reported in lines 404-407. The box plots represent median and interquartile range of the stained proteins. No cut-off has been used in this analysis. We have now further explained these aspects in the legend of Figure 2.

This manuscript is a resubmission of an earlier submission. The following is a list of the peer review reports and author responses from that submission.

Round 1

Reviewer 1 Report

This paper characterizes the JAK/STAT signaling pathway in various T-cell lymphomas and asks whether activation correlates with STAT3 and RHOA gene mutations, STAT3 phosphorylation levels, EBV seropositivity, immunohistochemical assessment of T-cell marker expression and clinical outcome. Cell line and drug sensitivity data is also included according to STAT3 phosphrylation status.

There are a number of problematic issues. Various end points and associations are made although none are robust. Hypotheses are tested in small cohorts of patients which are not well characterized and the subtypes of lymphomas and treatments differ for meaningful comparisons. The small sample size also detracts from making meaningful conclusions about therapeutic choices and clinical outcome.

Author Response

# We appreciate the comments raised by Reviewer 1. The study cohort has been essentially increased with 18 new samples increasing the number of study patients from 25 to 30 angioimmunoblastic lymphomas, from 8 to 13 ALK- and 5 to 8 ALK+ anaplastic large cell lymphomas, 8 to 12 PTL not otherwise specified and 3 to 4 control lymph nodes. Lymphoma samples have been sequenced for mutations in genes STAT3, RHOA, JAK1 and JAK3.  In addition, initial 49 samples have also been re-sequenced for JAK1 and JAK3 mutations. Unfortunately, we were not able to include SOCS1/3 in the sequencing due to technical problems.

The main objective of the manuscript was to investigate possible associations between STAT3, MAPK and AKT phosphorylation and PTCL histology and STAT3 and RHOA mutations. We believe that our study cohort is robust enough to answer these questions. To support these findings, we have also included data on the association of tumor CD30 phenotype, EBV positivity, sample type and sample origin with STAT3, MAPK and AKT phosphorylation. We believe the manuscript to be essentially improved and merit an additional review.

After revising the manuscript, we discovered that STAT3 phosphorylation was primarily associated with ALK+ and JAK1/STAT3-mutated and CD30+ ALK- ALCLs, thus, linking mutations in the JAK/STAT pathway to pathway activation.

Related to survival analyses, RHOA, pMAPK and pSTAT3 expression have been stained with IHC. To maximize patient numbers in survival analyses, we included all patients with AITL and PTCL NOS, which have been observed to share similar prognosis. Expression values were studied with multivariate Cox regression analysis using CHOP therapy as adjusting covariate. Hence, RHOA mutation, high pMAPK and high pSTAT3 markers were observed to be associated with poor survival independently of treatment.

However, in this study, 70% of AITL patients and 17% of PTCL NOS patients harbored a RHOA mutation. The prognostic impact of the mutation status, pMAPK % and pSTAT3 % therefore reflects primarily the subtype. When predicting overall survival after adjusting for treatment intensity (CHOP-treated vs. non-CHOP-treated), positive RHOA mutation status conferred a 2.58-fold hazard ratio (p = 0.019, CI95% [1.17-5.66]) while AITL subtype increased mortality risk by 2.78-fold (p=0.031 CI95% [1.10-7.06]) over PTCL NOS subtype. When both covariates are included in the same multivariate model, RHOA mutation status HR 1.90 p=0.17 [0.77-4.71] and AITL subtype HR 1.92 p=0.23 [0.65-5.64] implying that the mutation status reflects essentially the subtype and not independent risk from the mutation. Therefore, all survival analyses have now been omitted from the revised manuscript.

Reviewer 2 Report

What type of specimens were in the FFPE cases for IHC detection of pSTAT3, etc? As you know phosphorylation is labile and as formalin penetrates tissue at around 1mm per hour, large biopsies and excisions may no longer stain for pSTAT3 and others. This information is worth considering and possibly worth a supplemental table. In our experience we find that these IHC work well in core biopsies but not in excisions. Are presence of RHOA and high pMAPK independent predictors of survival? Or has this only taken into account treatment intensity? Worth clarifying.

Author Response

What type of specimens were in the FFPE cases for IHC detection of pSTAT3, etc? As you know phosphorylation is labile and as formalin penetrates tissue at around 1mm per hour, large biopsies and excisions may no longer stain for pSTAT3 and others. This information is worth considering and possibly worth a supplemental table. In our experience we find that these IHC work well in core biopsies but not in excisions.

# We thank reviewer 2 for insightful comments, which have significantly ameliorated the manuscript. FFPE specimens stained with IHC consisted mostly of enlarged lymph nodes objected for excision. Four samples were extracted from non-lymph node tissues and nine from biopsies. This information has been included in Supplementary Table 2. We did not observe general association with sample type or sample organ of origin and protein phosphorylation. However, higher pSTAT3 % was associated with biopsy samples (Supplementary Figure S4A). While biopsy samples were smaller in size than operated samples, sample size did not correlate significantly with pSTAT3 % (Supplementary Figure S4B). In addition, non-lymph node samples were associated with lower pAKT but no differences in pSTAT3 and pMAPK levels were observed (Supplementary Figure S4C). These results are also reported in lines 196-204.

Are presence of RHOA and high pMAPK independent predictors of survival? Or has this only taken into account treatment intensity? Worth clarifying.

# RHOA, pMAPK and pSTAT3 expression have been stained with IHC. To maximize patient numbers in survival analyses, we included all patients with AITL and PTL NOS, which have been observed to share similar prognosis. Expression values were studied with multivariate Cox regression analysis using CHOP therapy as adjusting covariate. Hence, RHOA mutation, high pMAPK and high pSTAT3 markers were observed to be associated with poor survival independently of treatment.

However in this study, 70% of AITL patient and 17% of PTCL NOS patients harbored a RHOA mutation. The prognostic impact of the mutation status, pMAPK % and pSTAT3 % therefore reflects primarily the subtype. When predicting overall survival after adjusting for treatment intensity (CHOP-treated vs. non-CHOP-treated), positive RHOA mutation status conferred a 2.58-fold hazard ratio (p = 0.019, CI95% [1.17-5.66]) while AITL subtype increased mortality risk by 2.78-fold (p=0.031 CI95% [1.10-7.06]) over PTCL NOS subtype. When both covariates are included in the same multivariate model, RHOA mutation status HR 1.90 p=0.17 [0.77-4.71] and AITL subtype HR 1.92 p=0.23 [0.65-5.64] implying that the mutation status reflects essentially the subtype and not independent risk from the mutation. Therefore, all survival analyses have now been omitted from the revised manuscript.

Reviewer 3 Report

Andersson et al. describe the expression of STAT3 with respect to STAT3, STAT5B and RHOA mutations in mature T-cell lymphomas and show that STAT3 is overexpressed/phosphorylated in many instances, which was independent of STA3 gene mutations. This is not completely new knowledge and additional work is need to complete the picture, i.e. studies of up-stream mechanisms as suggested below.

Major reservations:

My major reservation is the extremely low number of PTCL, NOS (only 8) and the rather low number of ALK+ ALCL (only 5) as well as ALK- ALCL (only 8). These numbers should be doubled, especially as the authors look for survival analysis, otherwise the survival data should be removed since not relevant. The PTCL are poorly characterized, I miss important data on the expression of GATA3 (Th2-like) and Tbet (Th1-like); see PMID:31562134. Since mutations of TET2, IDH2, DNMT3A on the one side, and - on the other and very relevant for the current paper - JAK1-3, SOCS1 and SOCS3 mutations and translocations are relevant in the studied entities (PMID:24413734, rev. in PMID:28182501, PMID:19136931), targeted resequencing of the already isolated DNA in that respect would highly increase the impact of the current study. In addition, since EBV may be linked to high pSTAT3 presence, all PTCL cases must be studied for EBV association (PMID:17997602) All cases should be stained for CD30 - as its expression and related signaling may explain STAT3 phosphorylation in the absence of mutations. Lines 134-136: the respective data should be also compared to the STAT5 Tyr705 phosphorylation. Lines 149-150: the VAFs should be put into the context of tumor cell amount in the respective cases since in many T-cell lymphomas it is hardly higher than 15% and the VAFs may simply reflect this fact. Given the fact that RHOA mutations are most commonly present in AITL, did the authors test their survival data if the mutation was indeed an independent prognostic factor or simply reflected AITL. Given the low number of cases in the study, the statements in lines 282-297 may be tempered or omitted.

Minor comments:

Gene names must be italicized throughout; this is not the case in 80% of the instances. Lines 193-195: the t-test comparison is very cumbersomely presented. Please revise.

Author Response

My major reservation is the extremely low number of PTCL, NOS (only 8) and the rather low number of ALK+ ALCL (only 5) as well as ALK- ALCL (only 8). These numbers should be doubled, especially as the authors look for survival analysis, otherwise the survival data should be removed since not relevant.

# We thank reviewer 3 for excellent feedback, which we believe helped to improve the manuscript. The study cohort has been essentially increased with 18 new samples increasing the total number of study patients from 25 to 30 angioimmunoblastic lymphomas, from 8 to 13 ALK- and 5 to 8 ALK+ anaplastic large cell lymphomas, 8 to 12 PTL not otherwise specified and 3 to 4 control lymph nodes. As detailed in comments to reviewer 1 and 2, we have removed survival analysis from the revised manuscript.

The PTCL are poorly characterized, I miss important data on the expression of GATA3 (Th2-like) and Tbet (Th1-like); see PMID:31562134. Since mutations of TET2, IDH2, DNMT3A on the one side, and - on the other and very relevant for the current paper - JAK1-3, SOCS1 and SOCS3 mutations and translocations are relevant in the studied entities (PMID:24413734, rev. in PMID:28182501, PMID:19136931), targeted resequencing of the already isolated DNA in that respect would highly increase the impact of the current study.

# We appreciate the request for more precise disease characterization, but unfortunately high-quality RNA could not be extracted from these FFPE samples and no frozen vials were available. We mentioned this important perspective in line 305-307. As suggested by the reviewer, samples have now also been sequenced for mutations in JAK1 and JAK3 genes. Only two patients with JAK1 mutations were identified, but none was observed to harbor JAK3 mutations. Unfortunately, we were not able to include SOCS1/3 in the sequencing due to technical problems.

In addition, since EBV may be linked to high pSTAT3 presence, all PTCL cases must be studied for EBV association (PMID:17997602)

# As asked, we studied for association of EBV and phosphorylation of STAT3, MAPK and AKT (Supplemental Figure 3B and lines 193-195). No association between EBV positivity and pSTAT3 and pMAPK expression was noted.

All cases should be stained for CD30 - as its expression and related signaling may explain STAT3 phosphorylation in the absence of mutations.

# Reviewer 3 also requested for comparison of tumor CD30 phenotype and pSTAT3 expression. As reviewer 3 predicted, positive CD30 phenotype was associated with higher pSTAT3 expression (Supplemental Figure 3A). CD30 positivity trended also for higher pAKT but no association with pMAPK was observed (Supplemental Figure 3A). However, highest STAT3 phosphorylation was detected in CD30+ subjects with either STAT3 or JAK1 mutations (Figure 4C). These corresponded mostly to ALK- ALCL subtype (6/8 patients; Figure 4D). In total, 6/12 of CD30+ ALK- ALCL patients harbored a mutation in JAK1/STAT3. Unexpectedly, lowest STAT3 phosphorylation was observed in CD30- patients with STAT3 mutations, which might be explained by non-activating mutations or additional block in the JAK/STAT pathway. These findings crucially changed the results and interpretation of the manuscript. Therefore, the title has been changed as well as lines 38-41, 170-183, 230-233, 276-282.

Lines 134-136: the respective data should be also compared to the STAT5 Tyr705 phosphorylation.

# Reviewer 3 asked to analyze STAT5 Tyr705 phosphorylation. STAT5 Tyr705 was initially stained with IHC, but we could not validate stainings with a positive control, and therefore, these results have been omitted from the manuscript. Hence, no association for STAT5 has been performed with the CCLE dataset.

Lines 149-150: the VAFs should be put into the context of tumor cell amount in the respective cases since in many T-cell lymphomas it is hardly higher than 15% and the VAFs may simply reflect this fact.

# Reviewer 3 asked to investigate for any link between STAT3 mutation VAF and STAT3 phosphorylation after observing tumor proportion. No association between mutation VAF and pSTAT3 could be noted in the total sample cohort after observing tumor cell proportion (added to line 154-156). Higher JAK1/STAT3 VAF trended for higher STAT3 phosphorylation but the result remained non-significant possibly due to sample size (81.3% vs. 61.1% mean pSTAT3 expression in tumors with high (n=4) vs. low (n=4) JAK1/STAT3 VAF p=0.22, t-test) as reported in lines 179-181.

Given the fact that RHOA mutations are most commonly present in AITL, did the authors test their survival data if the mutation was indeed an independent prognostic factor or simply reflected AITL.

# To exclude the finding that RHOA would simply reflect AITL subtype, reviewer 3 asked to test whether RHOA mutation would be a prognostic factor of PTCL subtype independent of subtype. In this study, 70% of AITL patient and 17% of PTCL NOS patients harbored a RHOA mutation. The prognostic impact of the mutation status, pMAPK % and pSTAT3 % therefore reflects primarily the subtype. When predicting overall survival after adjusting for treatment intensity (CHOP-treated vs. non-CHOP-treated), positive RHOA mutation status conferred a 2.58-fold hazard ratio (p = 0.019, CI95% [1.17-5.66]) while AITL subtype increased mortality risk by 2.78-fold (p=0.031 CI95% [1.10-7.06]) over PTCL NOS subtype. When both covariates are included in the same multivariate model, RHOA mutation status HR 1.90 p=0.17 [0.77-4.71] and AITL subtype HR 1.92 p=0.23 [0.65-5.64] implying that the mutation status reflects essentially the subtype and not independent risk from the mutation. Therefore, all survival analyses have now been omitted.

Given the low number of cases in the study, the statements in lines 282-297 may be tempered or omitted.

# We thank the reviewer for this notion. We have modified the text in lines 296-297 and 305-307.

Minor comments:

Gene names must be italicized throughout; this is not the case in 80% of the instances.

# We have reviewed the manuscript and corrected text formatting for gene names.

Lines 193-195: the t-test comparison is very cumbersomely presented. Please revise.

# The sentence has been simplified to clarify the result.